# Vitamin D Receptor (*VDR*) Gene Polymorphisms Modify the Response to Vitamin D Supplementation: A Systematic Review and Meta-Analysis

**DOI:** 10.3390/nu14020360

**Published:** 2022-01-15

**Authors:** Ricardo Usategui-Martín, Daniel-Antonio De Luis-Román, José María Fernández-Gómez, Marta Ruiz-Mambrilla, José-Luis Pérez-Castrillón

**Affiliations:** 1IOBA, University of Valladolid, 47011 Valladolid, Spain; 2Cooperative Health Network for Research (RETICS), Oftared, National Institute of Health Carlos III, ISCIII, 47011 Madrid, Spain; 3Department of Endocrinology, Clinical University Hospital, 47002 Valladolid, Spain; dluisro@saludcastillayleon.es; 4Department of Medicine, Faculty of Medicine, University of Valladolid, 47002 Valladolid, Spain; 5Instituto de Endocrinología y Nutrición (IENVA), University of Valladolid, 47002 Valladolid, Spain; 6Department of Cell Biology, Histology and Pharmacology, Faculty of Medicine, University of Valladolid, 47002 Valladolid, Spain; josefg@med.uva.es; 7Department of Surgery, Faculty of Medicine, University of Valladolid, 47002 Valladolid, Spain; martamaria.ruiz@uva.es; 8Department of Internal Medicine, Río Hortega University Hospital, 47002 Valladolid, Spain

**Keywords:** vitamin D receptor, VDR, vitamin D, polymorphisms, TaqI, FokI, vitamin D supplementation

## Abstract

The vitamin D receptor (VDR), a member of the nuclear receptor superfamily of transcriptional regulators, is crucial to calcitriol signalling. VDR is regulated by genetic and environmental factors and it is hypothesised that the response to vitamin D supplementation could be modulated by genetic variants in the *VDR* gene. The best studied polymorphisms in the *VDR* gene are Apal (rs7975232), BsmI (rs1544410), Taql (rs731236) and Fokl (rs10735810). We conducted a systematic review and meta-analysis to evaluate the response to vitamin D supplementation according to the BsmI, TaqI, ApaI and FokI polymorphisms. We included studies that analysed the relationship between the response to vitamin D supplementation and the genotypic distribution of these polymorphisms. We included eight studies that enrolled 1038 subjects. The results showed no significant association with the BsmI and ApaI polymorphisms (*p* = 0.081 and *p* = 0.63) and that the variant allele (Tt+tt) of the TaqI polymorphism and the FF genotype of the FokI variant were associated with a better response to vitamin D supplementation (*p* = 0.02 and *p* < 0.001). In conclusion, the TaqI and FokI polymorphisms could play a role in the modulation of the response to vitamin D supplementation, as they are associated with a better response to supplementation.

## 1. Introduction

The vitamin D receptor (VDR), a member of the nuclear receptor superfamily of transcriptional regulators, plays a crucial role in calcitriol or 1-alfa,25-dihidroxicolecalciferol (1α,25(OH)2D) signalling. VDR is activated by binding with 1α,25(OH)2D, which forms a heterodimer with the retinoid X receptor (RXR). The 1α,25(OH)2D-VDR-RXR complex migrates to the nucleus to regulate the transcription of genes involved in vitamin D effects including phosphorous and calcium metabolism, cell proliferation and the control of innate and adaptive immunity [1,2,3]. 

The *VDR* gene is located on chromosome 12 (12q13.11) and more than 900 allelic variants in the *VDR* locus have been reported. The best studied *VDR* gene polymorphisms are Apal (rs7975232), BsmI (rs1544410), Taql (rs731236) and Fokl (rs10735810). ApaI, TaqI and BsmI are silent genetic variants that increase mRNA stability. The FokI polymorphism is located on exon 2 and results in a protein shortened by three amino acids [4,5,6]. These genetic variants have been associated with a predisposition to chronic diseases such as type 2 diabetes, cancer, autoimmune diseases, cardiovascular alterations, rheumatic arthritis and metabolic bone diseases [7,8,9,10]. 

VDR regulation is determined by genetic and environmental factors [11]. The principal environmental factors associated with VDR regulation are diet, exposure to sunlight, infections and pollution [12,13,14,15]. It has been postulated that these environmental factors could modify vitamin D levels which regulate the receptor. The mechanism is not clearly understood but it is hypothesised that it may be through epigenetic mechanisms [16]. Other factors involved in VDR regulation are the intake of the vitamin D precursor and the production and activity of the ligand. Genetic factors could modulate the influence of environmental factors on VDR regulation [11]. In this scenario, it has been reported that the response to vitamin D supplementation differs widely between individuals and one hypothesis is that genetic variants in the *VDR* gene are important in the response to vitamin D supplementation. The polymorphisms in the *VDR* gene could modify the VDR activity and therefore could be the explanation for the different response to vitamin D supplementation [4,5,6,17]. Various authors have examined how genetic variants in the *VDR* gene are associated with the response to vitamin D supplementation, and the many genetic association studies show contradictory results [18,19,20,21]. Therefore, our objective was to conduct a systematic review and meta-analysis to evaluate the response to vitamin D supplementation according to the BsmI, TaqI, ApaI and FokI polymorphisms in the *VDR* gene. 

## 2. Material and Methods

### 2.1. Inclusion Criteria and Search Strategy

To analyse the influence of *VDR* genetic variants on the response to vitamin D supplementation, studies including serum vitamin D levels before and after supplementation according to the genetic distribution of the BsmI, TaqI, ApaI and FokI *VDR* polymorphisms were considered eligible for inclusion. 

This systematic review and meta-analysis were performed in accordance with the PRISMA guidelines [22] (Appendix A). We included studies evaluating the response to vitamin D supplementation according to genetic variants in the *VDR* gene. To identify eligible studies, we conducted a computer-based search in the PubMed, Web of Science, Scopus and Embase electronic databases up to November 2021. Potentially relevant articles were searched for using the following terms in combination with Medical Subject Headings (MeSH) terms and text words: “Vitamin D receptor”, “VDR”, “BsmI”, “TaqI”, “ApaI”, “FokI”, “polymorphism”, “mutations”, “variants”, “cholecalciferol”, “vitamin D”, “supplementation” and “vitamin D supplementation”. No language restrictions were applied. The references of selected articles were scanned to identify additional relevant articles. The MedLine option “related articles” and review articles on the topic were also used to supplement the search. 

### 2.2. Data Extraction

Bibliographic research and data extraction were conducted independently by three investigators (RUM, DDLR and JMFG). Differences were resolved by consensus with the senior author (JLPC). We extracted the authors names, the publication year, demographic information (age and sex), the follow-up time after vitamin D supplementation and serum vitamin D levels before and after supplementation according to the *VDR* gene polymorphisms. 

### 2.3. Statistical Analysis

Independent meta-analyses were carried out to compare baseline and post-supplementation serum vitamin D levels according to the genetic distribution of the *VDR* polymorphisms included. Sub-analyses by age and sex were also carried out. Meta-analysis was only carried out when ≥3 studies were available. We analysed all polymorphisms under a dominant model for the minor alleles. 

As previously described [23,24,25], meta-analyses were carried out using RevMan 5.0 software [26]. The difference between baseline and post-supplementation status and their 95% confidence interval (CI) were estimated for each study. Random-effects model was used to calculate the *p*-values (DerSimonian and Laird method). A *p*-value < 0.05 was considered statistically significant. To analyse the heterogeneity of the studies we applied Cochran’s Q-statistic (*p* < 0.10 indicated heterogeneity across studies). Inconsistency in the meta-analysis was estimated using the I^2^ statistic and this represented the percentage of the observed between-study variability due to heterogeneity. The following cut-off points were applied: (I^2^ = 0–25%, no heterogeneity; I^2^ = 25–50%, moderate heterogeneity; I^2^ = 50–75%, large heterogeneity; I^2^ = 75–100%, extreme heterogeneity). To assess publication bias, Begger’s funnel plot was examined based on visual inspection. Asymmetry suggested publication bias. Finally, sensitivity analyses to examine the effect of excluding individual studies were carried out. 

## 3. Results

### 3.1. Identification and Selection of Relevant Studies

Figure 1 shows the flow chart of the studies selected for inclusion in the meta-analysis. We initially identified 215 candidate articles for inclusion. After removing duplicates, the abstracts of 131 articles were reviewed and 103 were excluded. Thus, a total of 28 full text studies were assessed for eligibility. Of these, 20 articles were excluded because they did not contain the necessary information to carry out the meta-analysis (Appendix A). Therefore, eight studies that fulfilled the inclusion criteria were finally included in the meta-analysis [20,27,28,29,30,31,32,33]. The response to vitamin D supplementation according to the BsmI polymorphism in the *VDR* gene was analysed in six studies [20,27,28,29,30,32]. Five studies analysed the vitamin D response according to the genotypic distribution of the TaqI genetic variant [27,28,29,30,32]. The influence of the ApaI polymorphism was studied in four articles [27,29,30,32]. Finally, the influence of the FokI polymorphism in the response to vitamin D supplementation was analysed in five studies [27,29,30,31,33]. 

### 3.2. Study Characteristics

The studies included in the meta-analysis enrolled 1038 subjects. Detailed demographic characteristics are shown in Table 1. The mean age of the subjects included was 36.1 (10.2) years with a range of 10 to 78 years. Two studies included subjects aged <18 years [28,33] and one study only specified that subjects were aged >18 years [27]. There was a higher prevalence of women than men (77.8% vs. 8.6%). One article did not report the sex of the subjects [31]. The mean follow-up time after vitamin D supplementation was 7.4 (4.9) months. Baseline and post-supplementation serum vitamin D levels according to the BsmI, TaqI, ApaI and FokI polymorphisms in the *VDR* gene are summarized in Table 2. In the case of BsmI polymorphism, two studies associated the variant genotype with better response to vitamin D supplementation [27,30], two studies with worse response [20,32] and two studies did not show statistically significant association [28,29]. Five studies statistically associated the variant genotype of TaqI polymorphism with response to vitamin D supplementation [27,28,29,30,32]. Two studies associated the genotypic distribution of ApaI polymorphism with the response to supplementation [29,32]. For the FokI polymorphism, four articles showed association with response to vitamin D supplementation [27,30,31,33]. All studies used genomic DNA extracted from nucleated peripheral blood cells, and genotyping was performed using polymerase chain reaction-restriction fragment length polymorphism (PCR-RFLP).

### 3.3. Meta-Analysis of the Association between Gene Variants in the VDR Gene and the Response to Vitamin D Supplementation

The results of the meta-analysis are shown in Figure 2. The results showed that the BsmI genetic variant was not significantly associated with the response to vitamin D supplementation (*p* = 0.81, Figure 2A). In the case of the TaqI polymorphism, the variant allele (Tt+tt genotype) was significantly associated with a better response to vitamin D supplementation (*p* = 0.02, Figure 2B). There was no significant association between the ApaI variant and the response to vitamin D supplementation (*p* = 0.63, Figure 2C). Finally, subjects carrying the FF genotype of the FokI polymorphism in the *VDR* gene responded better to vitamin D supplementation than subjects with the variant allele (Ff+ff) (*p* < 0.001, Figure 2D). 

When a meta-analysis includes fewer than 10 articles, the power of the test for funnel plot asymmetry is too low to distinguish the probability of real asymmetry [34]. Even so, we examined publication bias by visual inspection using Begger’s funnel plot (Appendix A) and it appeared to be symmetrical, although there was some uncertainty regarding the degree of symmetry. 

The results were not modified by excluding articles that included only subjects aged <18 years or only analysing articles including females. Sub-analyses on the basis of ethnicity could not be carried out because the selected articles did not include this information. After sensitivity analysis, the exclusion of individual studies did not alter the results.

## 4. Discussion

The relationship between genetic variants in the *VDR* gene and the response to vitamin D supplementation remains unclear. Thus, we carried out a systematic review and meta-analysis to evaluate the response to supplementation according to the genotype distribution of the BsmI, TaqI, ApaI and FokI polymorphisms in the *VDR* gene. The results showed that the variant allele of the TaqI polymorphism and the FF genotype of the FokI variant were associated with a better response to vitamin D supplementation. The BsmI and ApaI polymorphisms were not associated with the response to vitamin D supplementation. 

Calcitriol signalling is crucial in bone metabolism as it is involved in calcium absorption, parathormone secretion and, therefore, bone resorption and cellular differentiation. Vitamin D deficiency has been associated with bone metabolism alterations [35,36,37]. Therefore, vitamin D intake as a preventive nutritional treatment of osteoporosis plays an important role in improving health status [38,39], but the efficacy of supplementation varies widely between subjects [18,19,20]. One explanatory hypothesis is that genetic variants in *VDR* could modulate the response to vitamin D supplementation. Our results showed that carrying the variant allele of the TaqI polymorphism was associated with a better response to vitamin D supplementation. TaqI is a silent polymorphism located in the 3´ *VDR* gene region and has been associated with an increase in mRNA stability [4,5,6]. A previous meta-analysis associated the TaqI genetic variant with the risk of bone fracture [10]. This may be in line with our results, as the TaqI polymorphism may modify the response to vitamin D supplementation and thus could modify the risk of bone fracture. However, other factors besides vitamin D levels are involved in the susceptibility to bone fracture [40]. Our meta-analysis also associated the FF genotype of the FokI polymorphism with a better response to vitamin D supplementation. The FokI polymorphism is located on exon 2 and the F allele has been associated with the translation of a more active protein [17]. The greater activity of VDR could be associated with a better response to vitamin D supplementation. In addition, the F allele of the FokI genetic variant has also been associated with better calcium absorption, higher bone mineral density and a reduced risk of vertebral bone fractures [41,42,43,44]. Therefore, it seems clear that the F allele of the FokI polymorphism is associated with greater VDR activity, improving the response to vitamin D and calcium supplementation and being associated with the risk of bone fracture. Finally, we also performed sub analysis by age and sex due to it having been reported that vitamin D metabolism is affected by these factors [45,46]. Our results were not modified when analysing according to age and gender. In this sense, we hypothesise that differences in vitamin D absorption caused by sex and age are probably more notable in subjects with the same genotype, and as our sample is very heterogeneous we do not observe differences.

This study had some limitations. Firstly, a general limitation of meta-analyses of genetic association studies—contradictory results and heterogeneity in the studies included—is quite common and reflects the true genetic heterogeneity of the different samples or hidden stratification of the population. Only a small number of studies were eligible for inclusion in our study and there was a lack of information in some, so they could not be included in the meta-analysis. Furthermore, several of the studies had low sample sizes with wide variations. Finally, the exposure to sunlight is one of the environmental factors which is crucial in VDR regulation [11]. Thus, it could have been interesting to analyse the results obtained as a function of sunlight exposure, but this could not be done because only one included paper reported this information [32]. Even with these limitations, this meta-analysis contributes significantly to our understanding the crucial role of *VDR* gene polymorphisms in the modulation of the vitamin D supplementation response. 

## 5. Conclusions

In conclusion, this meta-analysis advances our current understating of how *VDR* gene polymorphisms influence the response to vitamin D supplementation, providing moderate evidence that the variant allele of the TaqI polymorphism and the FF genotype of the FokI genetic variant were associated with a better response to vitamin D supplementation. Further research with a homogeneous design should be carried out to improve understanding of the role of *VDR* gene polymorphisms in the modulation of the response to vitamin D supplementation, and its possible clinical value.

## Figures and Tables

**Figure 1 nutrients-14-00360-f001:**
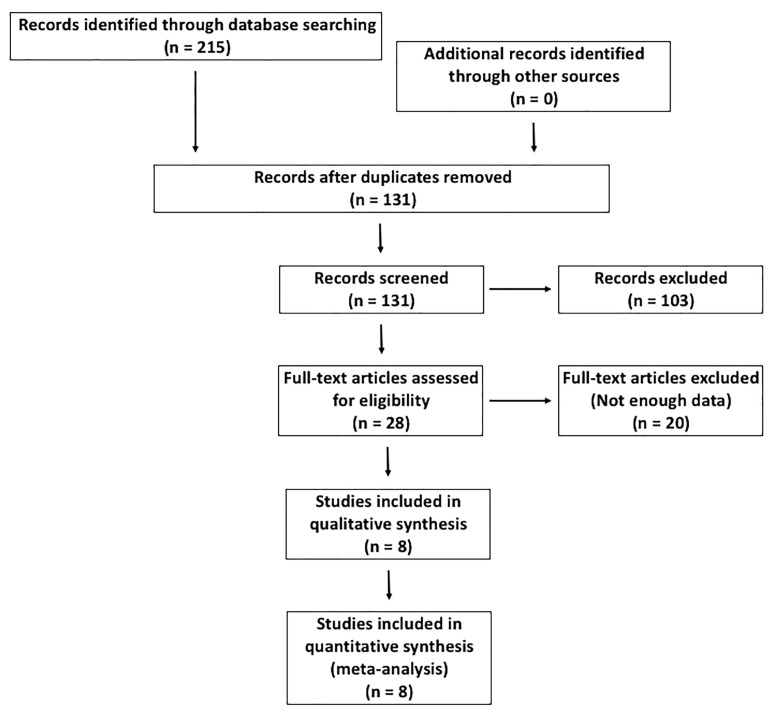
Flow chart of the studies selected for inclusion in the meta-analysis.

**Figure 2 nutrients-14-00360-f002:**
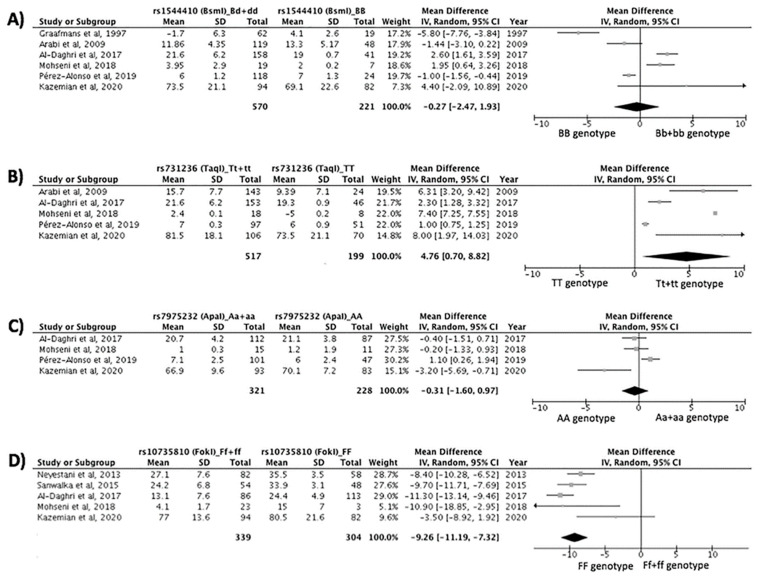
Meta-analysis of the association between gene variants in the vitamin D receptor (*VDR)* gene and the response to vitamin D supplementation. (**A**) Association between the BsmI polymorphism and the response to vitamin D supplementation. Test for overall effect: Z = 0.24 (*p* = 0.81). Test for heterogeneity: χ2 = 6.31 (*p* < 0.001), I2 = 9.4%. (**B**) Association between the TaqI polymorphism and the response to vitamin D supplementation. Test for overall effect: Z = 2.30 (*p* = 0.02). Test for heterogeneity: χ2 = 19.47 (*p* < 0.001), I2 = 10%. (**C**) Association between the ApaI polymorphism and the response to vitamin D supplementation. Test for overall effect: Z = 0.48 (*p* = 0.63). Test for heterogeneity: χ2 = 1.24 (*p* = 0.004), I2 = 7.7%. (**D**) Association between the FokI polymorphism and the response to vitamin D supplementation. Test for overall effect: Z = 9.39 (*p* < 0.001). Test for heterogeneity: χ2 = 2.47 (*p* = 0.04), I2 = 5.9%.

**Table 1 nutrients-14-00360-t001:** Characteristics of the studies included in the meta-analysis.

Authors, Year	N	Age [Years (SD)]	Gender [n (%)]		Country	Vitamin D Dose	Follow-Up Time
			Women	Men			
Graafmans et al., 1997	81	78 (5)	81 (100%)	0 (0%)	Netherlands	400 IU/24 h	12 months
Arabi et al., 2009	167	10 to 17	167 (100%)	0 (0%)	Lebanon	1100 IU/24 h	12 months
Neyestani et al., 2013	140	29 to 67	-	-	Iran	1000 IU/24 h	3 months
Sanwalka et al., 2015	102	11.2 (0.5)	102 (100%)	0 (0%)	India	333 IU/24 h	12 months
Al-Daghri et al., 2017	199	>18	114 (57.2%)	90 (42.8%)	Saudi Arabia	2000 IU/24 h	12 months
Mohseni et al., 2018	26	47.7 (8.0)	26 (100%)	0 (0%)	Iran	7000 IU/24 h	2 months
Pérez-Alonso et al., 2019	142	55 (4)	142 (100%)	0 (0%)	Spain	800 IU/24 h	3 months
Kazemian et al., 2020	176	48.6 (8.7)	176 (100%)	0 (0%)	Iran	4000 IU/24 h	3 months

SD: standard deviation, IU: international units.

**Table 2 nutrients-14-00360-t002:** Baseline and post-supplementation vitamin D levels according to the BsmI, TaqI, ApaI and FokI polymorphisms in the vitamin D receptor (*VDR*) gene.

Authors, Year	Vitamin D Levels BEFORE Supplementation, ng/mL [Mean (SD)]								Vitamin D Levels AFTER Supplementation, ng/mL [Mean (SD)]							
	rs1544410 (BsmI)		rs731236 (TaqI)		rs7975232 (ApaI)		rs10735810 (FokI)		rs1544410 (BsmI)		rs731236 (TaqI)		rs7975232 (ApaI)		rs10735810 (FokI)	
	BB	Bd+dd	TT	Tt+tt	AA	Aa+aa	FF	Ff+ff	BB	Bd+dd	TT	Tt+tt	AA	Aa+aa	FF	Ff+ff
Graafmans et al., 1997	26 (7.5)	29.2 (8.5)	-	-	-	-	-	-	30.1 (10.1)	25.75 (14.8)	-	-	-	-	-	-
Arabi et al., 2009	14.3 (9.4)	14.25 (7.9)	14.0 (8.5)	13.9 (7.7)	-	-	-	-	27.64 (14.5)	26.11 (12.3)	23.39 (15.6)	29.64 (15.5)	-	-	-	-
Neyestani et al., 2013	-	-	-	-	-	-	38.1 (21.5)	37.9 (16.7)	-	-	-	-	-	-	73.6 (25)	65 (24.3)
Sanwalka et al., 2015	-	-	-	-	-	-	27.77 (3.1)	22.8 (2.04)	-	-	-	-	-	-	61.72 (6.2)	47.02 (8.9)
Al-Daghri et al., 2017	31.1 (14)	34 (11.1)	31.9 (12.7)	33.8 (11.6)	35.1 (9.5)	33.3 (12.4)	33 (12.4)	34.8 (11.1)	50.1 (14.7)	55.6 (17.3)	51.2 (13.6)	55.4 (17.8)	56.2 (13.3)	54 (18.2)	57.4 (17.3)	47.9 (13.8)
Mohseni et al., 2018	9.0 (1.4)	12.75 (1.4)	16.5 (4.6)	12.2 (1.5)	13.6 (1.3)	13.6 (2.7)	13.0 (1.0)	11.2 (1.4)	11.0 (1.4)	16.7 (4.3)	11.5 (1.2)	14.6 (1.5)	14.8 (3.2)	14.6 (3.1)	28.0 (12)	15.3 (3.1)
Pérez-Alonso et al., 2019	21 (10)	24.5 (9)	25 (9)	23 (9.5)	23 (10)	24 (9)	-	-	28 (9)	30.5 (10)	31 (8)	30 (9)	29 (9.5)	31.1 (9.5)	-	-
Kazemian et al., 2020	30.2 (11.4)	41.7 (16.9)	31.8 (10.4)	37.4 (11.3)	40.9 (14.2)	31.35 (11.4)	34.4 (12.4)	30.8 (9.4)	99.3 (34)	131.2 (29)	105.3 (31.5)	118.9 (29.4)	111 (21.4)	98.3 (23)	114.9 (34)	107.8 (23)

SD: standard deviation.

## Data Availability

All data and results are available in this manuscript.

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
