# Peer review of "Vitamin D Receptor (VDR) Gene Polymorphisms Modify the Response to Vitamin D Supplementation: A Systematic Review and Meta-Analysis"

_nutrients, 2022, doi:10.3390/nu14020360_

Round 1

Reviewer 1 Report

The authors aimed to meta-analysis analyze the relationship between 2- 12 month vitamin D supplementation and eight studies of polymorphisms in the VDR gene are Apal (rs7975232), BsmI (rs1544410), Taql (rs731236) and Fokl (rs10735810).

The manuscript is largely well written and informative overall. However, there seem to be several  minor concerns in this manuscript.

Main remarks:

-The discussion lacked information on the influence of the interaction of insolation and the influence of the TaqI, FokI polymorphism. although in the introduction the authors stated that sunlight has a regulating effect on the VDR gene. The 7 articles for which the data were analyzed referred to rather sunny areas. Only the article by Graafmans et al, 1997 dealt with areas with moderate sunshine (Netherlands).

-Table 2 does not include the dose supplemented with Vit D

- Line 156 is " χ2=6.31 (P < 0.001)", it should be χ2 = 6.31 (p <0.001). Similarly, in the following lines: 158, 159, 161.

Author Response

Reviewer 1 comments

The authors aimed to meta-analysis analyze the relationship between 2-12 months’ vitamin D supplementation and eight studies of polymorphisms in the VDR gene are Apal (rs7975232), BsmI (rs1544410), Taql (rs731236) and Fokl (rs10735810). The manuscript is largely well written and informative overall. However, there seem to be several minor concerns in this manuscript.

We thank the reviewer for the comments and for the detailed revision of the manuscript. We believe that revisions based on your recommendations have improved the value and accuracy of our manuscript

The discussion lacked information on the influence of the interaction of insolation and the influence of the TaqI, FokI polymorphism. although in the introduction the authors stated that sunlight has a regulating effect on the VDR gene. The 7 articles for which the data were analyzed referred to rather sunny areas. Only the article by Graafmans et al, 1997 dealt with areas with moderate sunshine (Netherlands).

The exposure to sunlight in one of the environmental factors which is crucial in the VDR regulation (Saccone et al, 2015). It could have been interesting to analyse the results obtained as a function of sunlight exposure, but this could not be done because only one included paper reported this information (Pérez-Alonso M et al, 2020). We have included this in the Discussion section as a limitation of our study.

Table 1 does not include the dose supplemented with Vit D

We agree with the reviewer that this is a relevant point; we have included this information in the Table 1.

Line 156 is " χ2=6.31 (P < 0.001)", it should be χ2 = 6.31 (p <0.001). Similarly, in the following lines: 158, 159, 161.

Thank you very much for the comment, we have modified it.

Reviewer 2 Report

Vitamin D Receptor (VDR) gene polymorphisms modify the response to vitamin D supplementation: A systematic review and meta-analysis

I would like to thank the authors for the opportunity to review this text.

I found the article interesting, however some points that I think that should be address:

Was this review registered on PROSPERO?

I find that the introduction should better explore the relationship between VDR polymorphisms and vitamin D levels in response to vitamin D supplementation. Is there any hypothesis, even if theoretical, to explain why VDR polymorphisms influence vitamin D levels?

I would suggest a more complete introduction citing preexisting data on the subject.

The methods are too succinct. A meta-analysis is a complex statistical technic and further details should be provided. Please look at the Desideratum in the word document.

A qualitative synthesis is also lacking, describing in summary the main results of the studies included in the analyses.

Finally, I think that the practical significance and application of the findings should be explored.

Author Response

Reviewer 2 comments

I would like to thank the authors for the opportunity to review this text. I found the article interesting, however some points that I think that should be address.

We thank the reviewer for the comments and for the detailed revision of the manuscript. We believe that revisions based on your recommendations have improved the value and accuracy of our manuscript.

Was this review registered on PROSPERO?

We have already applied for registration with PROSPERO. Attached is a screenshot of the PROSPERO website.

I find that the introduction should better explore the relationship between VDR polymorphisms and vitamin D levels in response to vitamin D supplementation. Is there any hypothesis, even if theoretical, to explain why VDR polymorphisms influence vitamin D levels? I would suggest a more complete introduction citing pre-existing data on the subject.

We thank the reviewer for raising our attention to this point, we have expanded on it in the introduction section.

 The methods are too succinct. A meta-analysis is a complex statistical technic and further details should be provided. Please look at the Desideratum in the word document.

We have expanded and modified the Statistical analysis section to provide more details.

 A qualitative synthesis is also lacking, describing in summary the main results of the studies included in the analyses.

We have included a qualitative description of the studies included in the Study characteristics section.

Finally, I think that the practical significance and application of the findings should be explored.

We agree with the reviewer that this is a relevant point. We have included in the Conclusion section: Further research with a homogeneous design should be carried out to improve understanding of the role of VDR gene polymorphisms in the modulation of the response to vitamin D supplementation, and its possible clinical value”.

Reviewer 3 Report

Usategui-Martin et al. conducted a systematic review and meta-analysis of four polymorphic variants: TaqI, BsmI, ApaI and FokI located in the VDR gene and analyzed their association with response to vitamin D supplementation. In the introduction, the authors describe the function of the VDR, the location of the variants analyzed, and the regulation of the vitamin D receptor. The material and methods chapter clearly describes the selection of studies for analysis and the statistical methods used. The results of the analysis showed that variant allele of TaqI polymorphism and FF genotype of FokI variant were associated with better response to vitamin D supplementation. The authors discuss the impact of significant variants in a broader context and point out the limitations of the study. 
In my opinion, a paragraph describing in more detail the effects of age, gender, and ethnicity on vitamin D metabolism would have greatly improved the discussion, as these factors differentiate the studies reviewed. 
Minor concerns: editing issues (different fonts, double spaces).

Author Response

Reviewer 3 comments

Usategui-Martin et al. conducted a systematic review and meta-analysis of four polymorphic variants: TaqI, BsmI, ApaI and FokI located in the VDR gene and analyzed their association with response to vitamin D supplementation. In the introduction, the authors describe the function of the VDR, the location of the variants analyzed, and the regulation of the vitamin D receptor. The material and methods chapter clearly describes the selection of studies for analysis and the statistical methods used. The results of the analysis showed that variant allele of TaqI polymorphism and FF genotype of FokI variant were associated with better response to vitamin D supplementation. The authors discuss the impact of significant variants in a broader context and point out the limitations of the study.

We thank the reviewer for the comments and for the detailed revision of the manuscript. We believe that revisions based on your recommendations have improved the value and accuracy of our manuscript

 In my opinion, a paragraph describing in more detail the effects of age, gender, and ethnicity on vitamin D metabolism would have greatly improved the discussion, as these factors differentiate the studies reviewed.

Our results were not modified by excluding articles that included only subjects aged <18 years or only analysing articles including females. Sub-analyses on the basis of ethnicity could not be carried out because the selected articles did not include this information. In this sense, we have added in the Discussion section: “Finally, we also performed sub analysis by age and sex due to it had been reported that vitamin D metabolism is affected by them (Gallagher et al, 2013 and Verdoia et al, 2015). Our results were not modified analysing according to the age and gender. In this sense, we hypothesise that differences in vitamin D absorption caused by sex and age are probably more notable in subjects with the same genotype and as our sample is very heterogeneous we do not observe differences”

Minor concerns: editing issues (different fonts, double spaces).

We thank the reviewer for raising our attention to this point, we have checked and unified the format.